# Effect of mitigation measures on the spreading of COVID-19 in hard-hit states in the U.S.

Ka-Ming Tam[1,2]◉*, Nicholas Walker[1]◉, Juana Moreno[1,2]◉

**1** Department of Physics & Astronomy, Louisiana State University, Baton Rouge, Louisiana, United States of America, **2** Center for Computation & Technology, Louisiana State University, Baton Rouge, Louisiana, United States of America

◉ These authors contributed equally to this work.
* kmtam@lsu.edu

## Abstract

State government-mandated social distancing measures have helped to slow the growth of the COVID-19 pandemic in the United States. Many of the current predictive models of the development of COVID-19, especially after mitigation efforts, partially rely on extrapolations from data collected in other countries. Since most states enacted stay-at-home orders towards the end of March, the resulting effects of social distancing should be reflected in the death and infection counts by the end of April. Using the data available through April 25th, we investigate the change in the infection rate due to the mitigation efforts and project death and infection counts through September 2020 for some of the most heavily impacted states: New York, New Jersey, Michigan, Massachusetts, Illinois, and Louisiana. We find that with the current mitigation efforts, five of those six states have reduced their base reproduction number to a value less than one, stopping the exponential growth of the pandemic. We also project different scenarios after the mitigation is relaxed.

**Data Availability Statement:** The raw data generated from our model is available via: https://github.com/kmtam/covid19_data The graphs generated from the raw data can be found via:

## Introduction

As of April 25, there are 2.8 million confirmed cases and close to 200,000 deaths attributed to COVID-19 in the world, with over 0.9 million cases and close to 52,000 deaths in the United States. The first confirmed case of COVID-19 in the United States occurred in Washington State on January 20, 2020. Until early March, the number of reported cases remained rather low, with most of them residing in the states of Washington, New York, and California. However, since early March, the disease has spread to all states, with both recorded deaths and infections growing at alarming rates in many states. Between mid-March and early-April, most states issued stay-at-home (SaH) orders.

Since the effects of social distancing measures in the United States have not be known until recently, most studies of the progress of the epidemic are largely based on extrapolations from the effects of similar strategies in other countries. However, by mid-April most states have shown signs of slowing the initial exponential growth of infection. Understanding the effects

https://covid19projection.org/ The source of the daily death and confirmed cases are obtained from a New York Times database: https://github.com/nytimes/covid-19-data.

**Funding:** This work is funded by the National Science Foundation EPSCoR CIMM project under award OIA-1541079. This work used the high performance computational resources provided by the Louisiana Optical Network Initiative and HPC@LSU computing. The funders had no role in study design, data collection and analysis, decision to publish, or preparation of the manuscript.

**Competing interests:** The authors have declared that no competing interests exist.

of mitigation efforts based on local data is important, as different countries have implemented different degrees of social distancing measures and their effects simply cannot be translated between countries. Moreover, understanding the effect of mitigation is key to predicting the effects of relaxing those efforts. How will the timing affect the number of infections? What measures need to be enforced to keep the infection rate sufficiently low to prevent exponential growth again? For these reasons, we expand our preliminary study of the early stage of COVID-19 epidemic in Louisiana. [1] As more data is available, we now estimate the death rate and recovery rate of those in quarantine, which allows us to predict the death count, positive confirmed count, and perhaps more importantly, the infected yet unidentified count after social distancing measures.

The goal of this study is to extract the dynamics of COVID-19 in some of the most heavily impacted states and to investigate the change of the infection rate after the effects of the stay-at-home orders. We then model several scenarios with different dates for the release of the stay-at-home orders and different hypothetical increases of the infection rate. We also compare our results to the widely publicized model by the Institute for Health Metrics and Evaluation (IHME). [2]

This paper is organized as follows. In the Model section, we present the model. In the Method section, we present the method for extracting the parameters of the model. In the Results section, we present the results for six states. These include New York, New Jersey, Michigan, Illinois, Massachusetts, and Louisiana. In the Discussion section, we discuss the error sources and the possible improvement of the projection. In the appendix, we benchmark our projection to that of the IHME.

## Model

We use the Susceptible-Infected-Recovered (SIR) model [3, 4] modified to consider the number of quarantined people. Similar modifications on the SIR model have been considered elsewhere to model the spread of COVID-19. [5–34] The equations defining the dynamics of the model are as follows:

$$\frac{dS(t)}{dt} = -\beta \frac{S(t)I(t)}{N}, \tag{1}$$

$$\frac{dI(t)}{dt} = \beta \frac{S(t)I(t)}{N} - (\alpha + \eta)I(t), \tag{2}$$

$$\frac{dQ(t)}{dt} = \eta I(t) - \delta(t)Q(t) - \xi(t)Q(t), \tag{3}$$

$$\frac{dR(t)}{dt} = \xi(t)Q(t) + \alpha I(t), \tag{4}$$

$$\frac{dC(t)}{dt} = \delta(t)Q(t), \tag{5}$$

where $N$ is the total population size, $S$ is the susceptible population count, $I$ is the unidentified while infectious population count, $Q$ is the number of identified positive cases which are quarantined, $R$ includes the number of recovered patients, and $C$ is the number of deaths. The model is characterized by the following parameters: $\beta$ is the infection rate, $\eta$ is the detection rate, $\alpha$ is the recovery rate of asymptomatic people, $\xi$ is the recovery rate of the quarantined patients, and $\delta$ is the casualty rate of the quarantined. All the parameters are in units of (1/

day). The quarantined population $Q$ is composed of the identified positive cases independently of whether they are hospitalized or at home. We further assume that all casualties had been in quarantine prior to death and we consider that only $\xi$ and $\delta$ have time dependence. All of these assumptions are approximations made to allow for inference of the model parameters from the current available data.

The total death count at time $t$, $D(t)$, can be estimated as:

$$D(t) = \int_0^t \frac{dC(\tau)}{d\tau} d\tau. \tag{6}$$

The confirmed positive count is $P(t) = Q(t) + R_Q(t) + C(t)$, where $R_Q(t)$ are the recovered patients previously in quarantine. $P(t)$ can be estimates as:

$$
\begin{aligned}
P(t) &= \int_0^t \frac{dP(\tau)}{d\tau} d\tau \\
&= \int_0^t \left( \frac{dQ(\tau)}{d\tau} + \frac{dR_Q(\tau)}{d\tau} + \frac{dC(\tau)}{d\tau} \right) d\tau,
\end{aligned}
\tag{7}
$$

$$= \int_0^t \eta I(\tau) d\tau. \tag{8}$$

## Method

We determine two sets of parameters, one before the stay-at-home order and the other after the social distancing measures are in place. The method for estimating the model parameters from the data prior to the stay-at-home orders have been discussed in our previous work. [1] We repeat our approach here for completeness of the present paper.

Adequate testing for COVID-19 remains limited in the USA. For this reason, accurately predicting the trajectory of the spread of COVID-19 by relying on the number of confirmed cases alone is a rather questionable approach, especially for early stages in which the percentage of people tested was very small and the spread from asymptomatic infected people was significant. Alternatively, the number of fatalities attributed to COVID-19 combined with the mortality rate may be a more reliable estimator of the dynamics of the virus spread. Therefore, we extract the dynamics of COVID-19 from the death counts supplemented by the number of confirmed cases. In countries with better testing capabilities, the number of cases might be a better predictor.

At the beginning of the epidemic, only a small fraction of the population is infected, so we can assume the susceptible population count is very close to that of the total population, $S \sim N$. With this assumption, one can decouple Eq 2 from the rest of the Eqs 1, 3, 4 and 5, which gives us,

$$\frac{dI(t)}{dt} = \beta \frac{S(t)I(t)}{N} - (\alpha + \eta)I(t) \approx [\beta - (\alpha + \eta)]I(t). \tag{9}$$

The above equation can be solved to obtain the unidentified infected population count in terms of model parameters: [5, 7]

$$I(t) \approx I(0) \exp\left[(\beta - (\alpha + \eta))t\right], \tag{10}$$

At the beginning of the virus spread, the number of quarantined patients is also small compared to the number of infected. With the assumption $Q \ll I$, we consider the number of

confirmed cases at the start of the epidemic, $Q(t)$.

$$\frac{dQ(t)}{dt} = \eta I(t) - \delta(t)Q(t) - \xi(t)Q(t) \approx \eta I(t). \tag{11}$$

We are able to simplify Eq 3 by substituting Eq 10 into Eq 11 to obtain:

$$Q(t) = \frac{\eta}{\beta - (\alpha + \eta)} I(t). \tag{12}$$

Combining Eqs 12 and 5, we relate the rate of increase in the number of casualties with the number of infected people in the *early stages* of the epidemic:

$$\frac{dC(t)}{dt} = \delta(t)\frac{\eta}{\beta - (\alpha + \eta)} I(t) = \delta_0 I(t), \tag{13}$$

where $\delta_0$ is the mortality rate. Finally, the casualty count can be obtained by solving the above equation:

$$C(t) = \frac{\delta_0 I(0)}{\beta - (\alpha + \eta)} \exp\left[(\beta - (\alpha + \eta))t\right]. \tag{14}$$

In the beginning of the epidemic, it is reasonable to assume exponential growth in the number of fatalities since the mechanisms for slowing the dynamics, such as improved detection and social distancing, are delayed in time. To find the initial exponent, $\beta - (\alpha + \eta)$, and the prefactor, the death count and the number of deaths per day are fit to Eq 14 and its derivative $\left(\frac{dC(t)}{dt}\right)$, with the first date with one death taken as $t = 0$. We perform a three-day moving average to smooth the data prior to the fit. We also discard data with less than ten deaths and use the data for the next ten days. All states in this study were still in the exponential growth phase at the last day used for the fitting of the exponent. To identify the initial number of infected people, $I(0)$, $\delta_0$ must be estimated.

The mortality rate, $\delta_0$, is estimated by combining the accumulated mortality rate data and the median time between infection and death. The median time between infection and the onset of symptoms is about five days while the median time between the onset of symptoms and death is eight days. [35–38] It is worth noting that the distribution of these time intervals is close to log-normal, thus a more sophisticated analysis should include the effects of the non-self-averaging behavior of the distribution. Only the median values are used in the present work.

The accumulated mortality rate is estimated to be 2.3%. [39] Notably, the mortality rate does indeed vary by region. This may be due to the rate of testing as well as the capacity of health care facilities. For areas in which hospitals have been overrun, the death rate would be much higher. Notwithstanding these uncertainties, assuming that the health care facilities have not yet been overrun, the mortality rate is estimated to be $\delta_0 \approx \frac{0.023}{5 + 8} \approx 0.0018/\text{day}$.

We also estimate the recovery rate of asymptomatic people, $\alpha$, based on our current knowledge of the epidemic. Assuming that the average time to recovery or death from infection are both 13 days and that half of the infected never show any symptoms, [40] we estimate $\alpha = 0.5/13 \approx 0.0385/\text{day}$. This is likely closer to an upper bound of the estimate, as this parameter could easily be smaller in reality.

We estimate $\eta$ by minimizing the $\chi^2$ of the total number of deaths and confirmed cases as well as their derivatives (daily number of deaths and daily number of new cases) for the last five days of the ten-day interval we are considering after the death count rises to ten deaths.

After obtaining $\eta$, we can also infer the infection rate $\beta$ and the reproduction number $R_0 \approx \beta/(\eta + \alpha)$. [7]

As opposed to our previous work, [1] in which less data was available, we also estimate both the death rate, $\delta(t)$, and the recovery rate of the quarantined, $\xi(t)$, from the raw data of confirmed cases and death count as a function of time. Based on the assumption that the average time from the onset of symptoms to death or recovery is eight days, $\delta(t) + \xi(t) = 1/8 = 0.125/$ day. For days between the fourth and the eleventh ($t \geq 4$ and $t \leq 11$), we assume:

$$\delta(t) \approx \frac{1}{8}\frac{D(t+4) - D(t-4)}{P(t-4) - D(t-4)}. \tag{15}$$

For day 12 and beyond ($t \geq 12$) we assume:

$$\delta(t) \approx \frac{1}{8}\frac{D(t+4) - D(t-4)}{P(t-4) - P(t-12) - D(t-4)}. \tag{16}$$

We assume $\delta$ for days 1 through 3 is equal to our estimate for day 4. In order to make projections, we also consider that for days in the future, the value of $\delta$ is equal to the value for the last dat with available data. This is not an unreasonable approximation, as we find the value of $\delta$ is more or less stable after the stay-at-home order becomes effective. Fig 1 displays the values of $\delta$ in Louisiana for days between March 16 and April 19. Estimates for other states behave in a similar manner.

Finally, we look at the effects of current mitigation efforts. Within the present model, there are two major routes for slowing the initial exponential growth of the epidemic, either to decrease the infection rate $\beta$ or to increase the testing rate $\eta$. Increasing the recovery rate of unidentified infectious people, $\alpha$, can also reduce the spread, but this is unlikely to be achieved. It is expected that the stay-at-home orders reduce the infection rate but do not influence the testing rate. However, the effect is not universal, but rather highly dependent on the measures

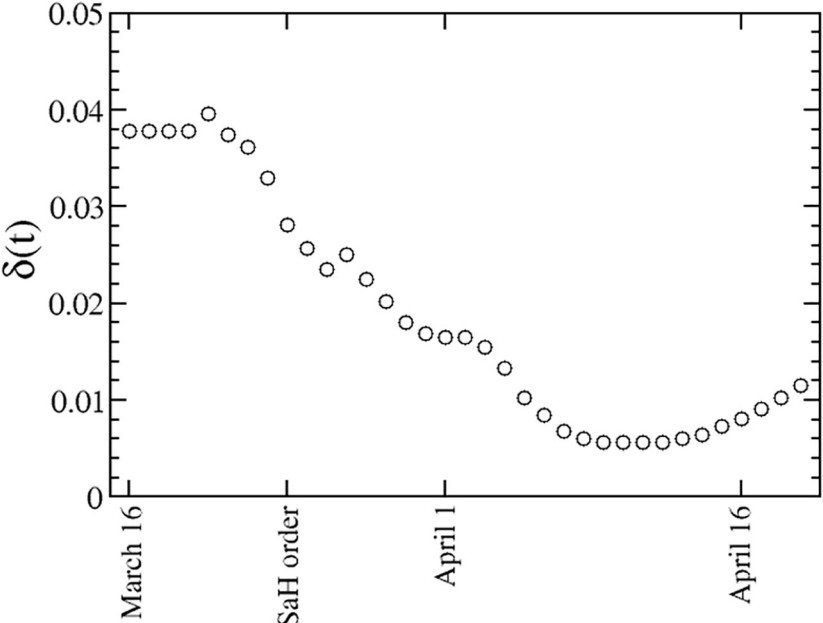

**Fig 1. Estimate of the casualty rate of the quarantined cases as a function of time, $\delta(t)$ in units of (1/day), for the state of Louisiana.**

imposed. Instead of extrapolating the data from other areas, we choose to determine it from the casualty and confirmed case counts. In addition, there is a time delay for the stay-at-home order to influence the number of cases and deaths. Therefore we consider the effects of social distancing measures to be reflected in two parameters, the reduction of the infection rate, $r$, and the first day when the measurements are effective, $d_r$. We determine both parameters by minimizing the $\chi^2$ of the values and daily changes of the death and the confirmed infected count for the five days between April 20 and 24.

Using $\delta(t)$ as calculated in Eqs 15 and 16 alongside the initial number of infected, $I(0)$, and the rest of parameters as shown in Table 1, we can solve the dynamics of the epidemic and estimate the death count, confirmed infected count, and perhaps more importantly the infected but unidentified count, $I(t)$, in each state.

## Results

### Results for intact mitigation efforts

We chose six states with high death counts to test our projections. These include New York (NY), New Jersey (NJ), Michigan (MI), Massachusetts (MA), Illinois (IL), and Louisiana (LA). The casualty and the confirmed case counts are obtained from the database of the New York Times. [41] Table 1 displays the parameters of our model for these states. In particular, we can compare the reproduction number in the exponential growth phase of the epidemic $R_0 \approx \frac{\beta}{\eta + \alpha}$ with an effective $R_0^{SaH} \approx \frac{r\beta}{\eta + \alpha}$ after the SaH order becomes effective. While the $R_0$ values are between 2.64 for Massachusetts and 4.83 for Louisiana, the $R_0^{SaH}$ values are between 1.01 for Illinois and 0.24 for Louisiana, showing that SaH orders have been effective to reduce $R_0$ to values less than one and control the exponential growth of the disease in most states.

We solve Eqs 1–8 and Fig 2 shows our predictions for the daily casualties ($C(t)$), total number of casualties ($D(t)$), and total number of confirmed cases ($P(t)$) for six states: New York, New Jersey, Michigan, Massachusetts, Illinois and Louisiana. Additionally, the casualty and confirmed case counts through April 25th that are used to find the model parameters are included in the plots. First, we see that all of the states have left the exponential phase and are flattening towards a quasi-linear region about one to two weeks after the SaH orders. However, the number of cases and fatalities in Illinois are still rapidly growing, albeit at a smaller rate than before.

**Table 1. Parameters for different states: The initial infection rate $\beta$, the detection rate $\eta$, the initial reproduction number $R_0 \approx \beta/(\eta + \alpha)$, the initial number of infected people on the day of the first confirmed death $I(0)$, the first date that social distancing measures are effectively reducing the infection rate in number of days since SaH order $d_r$, the current reduction in the infection rate $r$ as a proportion of the initial reproduction number, the reproduction number after SaH orders $R_0^{SaH}$, the day of the first death, and the date of the SaH order.**

| State | $\beta$ | $\eta$ | $R_0$ | $I(t=0)$ | $d_r(t)$ | $r$ | $R_0^{SaH}$ | $t=0$ | SaH(t) |
|---|---|---|---|---|---|---|---|---|---|
| NY | 0.484 | 0.070 | 4.46 | 1723 | 9(16) | 0.13 | 0.58 | 3/15 | 3/22(7) |
| NJ | 0.436 | 0.091 | 3.36 | 132 | 14(24) | 0.20 | 0.67 | 3/11 | 3/21(10) |
| MI | 0.449 | 0.057 | 4.69 | 881 | 8(13) | 0.11 | 0.52 | 3/19 | 3/24(5) |
| MA | 0.445 | 0.130 | 2.64 | 884 | 10(13) | 0.33 | 0.87 | 3/21 | 3/24(3) |
| IL | 0.421 | 0.107 | 2.89 | 481 | 11(14) | 0.35 | 1.01 | 3/18 | 3/21(3) |
| LA | 0.379 | 0.040 | 4.83 | 479 | 10(18) | 0.05 | 0.24 | 3/15 | 3/23(8) |

Massachusetts has not implemented a stay-at-home order but closed non-essential services on March 24th. The recovery rate of asymptomatic people $\alpha = 0.0385$ is assumed as a constant of the model. Note that the data for which the simulation begins is defined as $t = 0$ and the number in the parentheses for columns $d_r$ and SaH are the elapsed days from the simulation start.

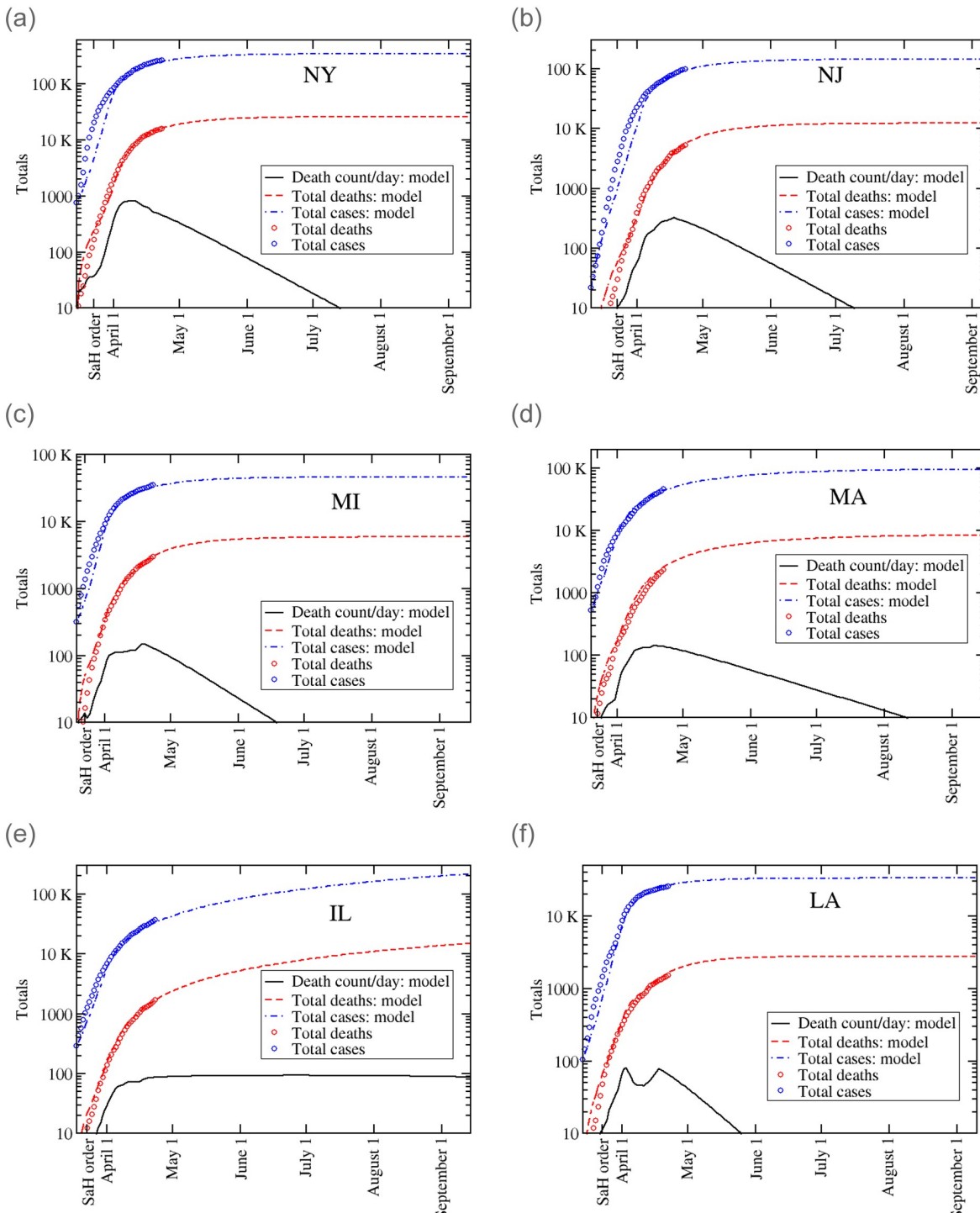

**Fig 2. Log-scale of the daily death count (black solid curve), total number of casualties (dashed red curve), and total number of confirmed cases (dash-dotted blue curve) as functions of time for six states: New York, New Jersey, Michigan, Massachusetts, Illinois and Louisiana.** The data for the number of deaths and cases are included as circles.

**Table 2. The total number of casualties and confirmed cases as of April 25 and projected total deaths and cases by September 1, in six states.**

| State | Current (4/25) deaths | Current (04/25) cases | Projected (9/1) deaths | Projected (9/1) cases |
|---|---|---|---|---|
| NY | 16,599 | 282,174 | 25,842 | 339,826 |
| NJ | 5,863 | 105,523 | 12,259 | 143,672 |
| MI | 3,274 | 37,203 | 5,882 | 45,801 |
| MA | 2,730 | 53,348 | 8,414 | 96,000 |
| IL | 1,884 | 41,777 | 13,746 | 199,474 |
| LA | 1,644 | 26,512 | 2,789 | 33,325 |

The results of the model agree reasonably well with the real casualty and case counts, in particular with the casualty counts used to estimate the initial exponent of the epidemic growth. The worst fits occurs with New York and New Jersey. Those are also the states with the largest number of cases. Difficulties in modeling New Jersey also appear in other models. [42] The issue might be related to the fact that NJ provides suburban housing for two large metropolitan areas, New York and Philadelphia. An analysis by metropolitan area instead of by state might be more meaningful in the case of New Jersey. Table 2 displays the current, as April 25, casualty and confirmed case counts alongside our projections through September 1.

It is worthwhile to compare our results with the widely used Institute for Health Metrics and Evaluation (IHME) model. [2] The projected total death counts of all six states we analyze are well within the 95% confidence interval of the IHME model on their update by Apr 25, except for Illinois. We emphasize that the present analysis is entirely based on the dynamical modeling of disease spreading with the necessary parameter inferred from death and confirmed counts alone. There is no extrapolation or interpolation of data from other countries or regions.

Unlike models based on statistical inference, the present model can provide additional information on the epidemic dynamics. We focus on further analyzing the predictions for Louisiana by plotting the full set of variables. In particular, this provides a hint of the number of infected but never identified cases. Consequently, the total number of infections can also be inferred. Fig 3 displays the daily death count $\left(\frac{dC(t)}{dt}\right)$, total number of casualties ($D(t)$), number of unidentified infected ($I(t)$), number of quarantined patients ($Q(t)$), total number of confirmed cases ($P(t)$), and total number of recovered people $\left(\int_0^t \frac{dR(\tau)}{d\tau}\,d\tau\right)$ as a function of time. Note that by September 1, the total number of recovered people (previously in quarantine or unidentified) is 62,509, almost double of the 33,325 projected confirmed cases.

Due to the underlying present model assumption that the mitigation efforts remain unchanged, we are implicitly assuming that the infection rate remains unchanged as well. Starting from late April to May, all states have planned reopening to a certain degree. It is expected the infection rate will be increased due to relaxation of the mitigation efforts. In the following subsection, we provide different scenarios of possible development after relaxing the mitigation effort.

## Results for relaxed mitigation efforts

The number of daily confirmed cases dropping below one per million within the population is a criterion for relaxing the social distancing measures for models based in statistical inference methods. [2] Since the present model considers the dynamics of the pandemic, we can estimate

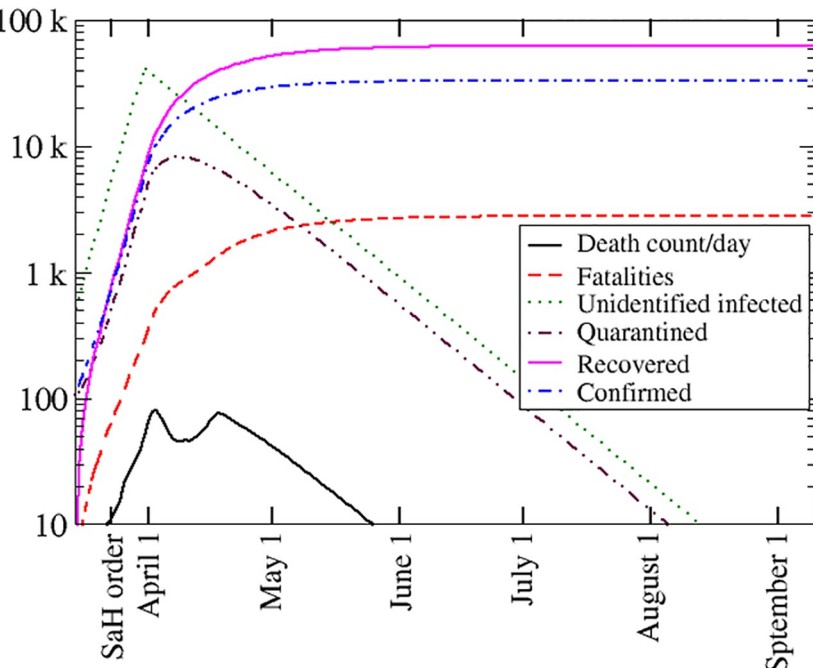

**Fig 3. Model predictions for Louisiana: Daily death count (solid black curve), total number of casualties (dashed red curve), number of unidentified infected (green dotted curve), count of quarantined patients (double-dot-dashed maroon curve), total number of confirmed cases (dot-dashed blue curve), and total number of recovered people (solid magenta curve) as functions of time.**

the increase in the infection and death count by proposing an increase in the infection rate due to relaxing the stay-at-home orders.

We explore possible scenarios after the relaxation of SaH orders for the state of Louisiana. We represent the effect of relaxing the mitigation efforts by increasing the infection rate $\beta$. Because the number of susceptible persons $S(t)$ has not changed sufficiently to reach herd immunity, if the value of $\beta$ reverts to the one before mitigation, the number of infected people will again grow exponentially. We investigate the effect of increasing $\beta$ at different times, e.g. May 1, May 16, and June 1. The extent to which $\beta$ will increase once SaH measures are relaxed depends on various factors, such as possible limitation of mass gatherings and the proportion of the population wearing personal protective gear. Figs 4 and 5 show predictions for the number of confirmed cased and fatalities, respectively, for different scenarios. We assume that the infection rate, $\beta$, increases to 25% and 50% of its value prior to the SaH order. We see that if $\beta$ increases to 25%, confirmed cases and deaths grow sub-exponentially but with a larger slope than the case with full mitigation efforts. If $\beta$ increases to 50%, both confirmed cases and fatalities will grow exponentially again. We notice that the delay on relaxing the mitigation does not substantially help to lower the number of infections in the long term.

## Discussion

We analyze the dynamics of COVID-19 spreading on six states. By late April, most states have been under stay-at-home orders for almost a month, and the effects of social distancing are reflected in the data. Additionally, we are able to estimate the recovery rate and the death rate for the patients under quarantine. This allows us to expand our previous study [1] and estimate the number of unidentified infected people, which is largely absent in models based on

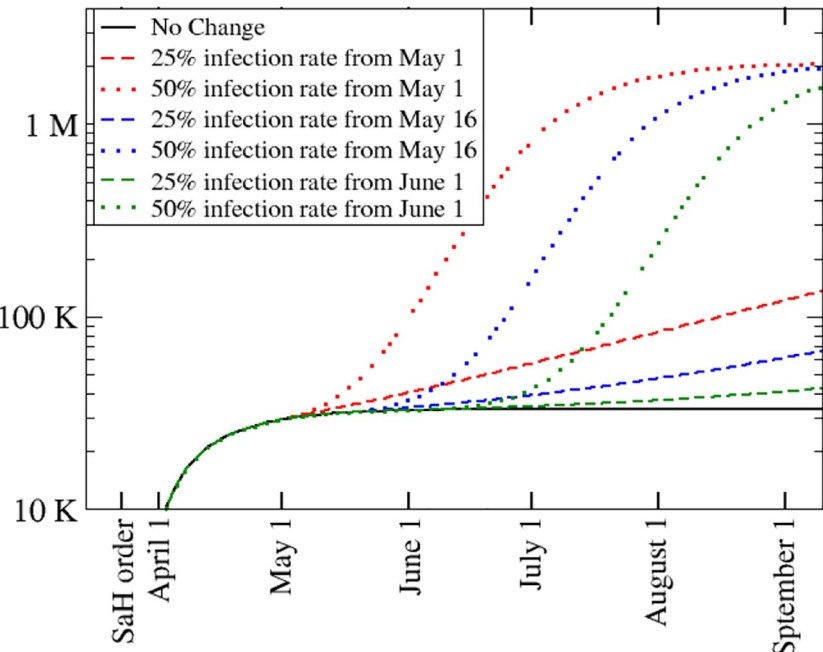

**Fig 4. Total number of cases as a function of time for several scenarios:** Full mitigation efforts are in place (solid black line), the infection rate, $\beta$, returns to 25% (dashed curves) and 50% (dotted curves) of its value prior to the SaH order at three different times, May 1, May 16, June 1.

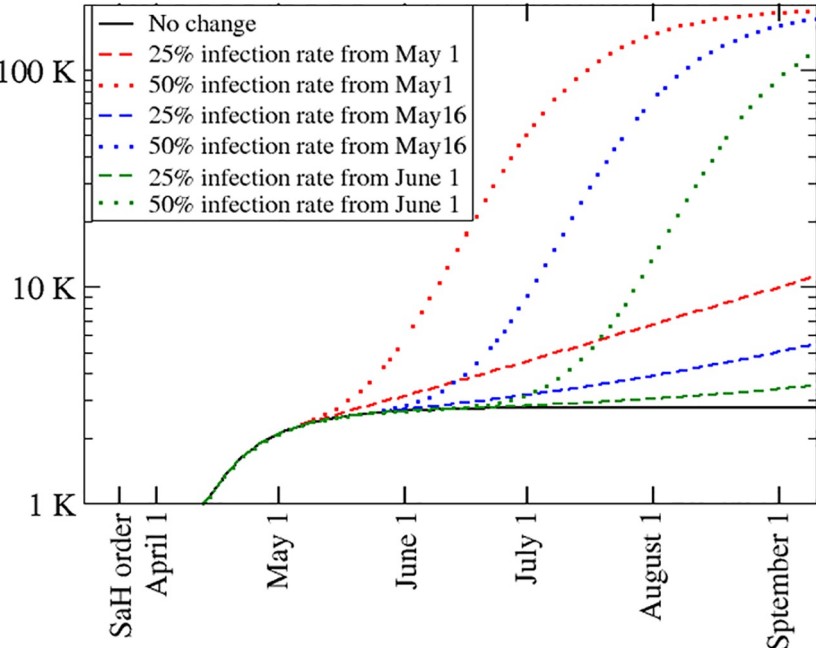

**Fig 5. Total number of death counts as a function of time for several scenarios:** Full mitigation efforts are in place (solid black line), the infection rate, $\beta$, returns to 25% (dashed curves) and 50% (dotted curves) of its value prior to the stay-at-home order at three different times, May 1, May 16 and June 1.

statistical inference. Our results confirm the widely believed speculation that the number of infected is much larger than the confirmed positive cases.

We find that by late April, the infection rate and the effective reproduction number in all the considered states have been significantly reduced. However, Illinois is still in the phase of increasing daily death and confirmed positive counts as of April 25th (see Table 1). Within this model, we do not assume that the change in the death count is exponentially decreasing after social distancing measures, but we use the current data to estimate the dynamics of the disease spreading.

We also provide possible scenarios of reopening with a focus in Louisiana. As there is no data available from the United States, a reasonable assumption is that the infection rate will increase after the SaH order is relaxed. If the infection rate returns to a value close to the one at the beginning of the epidemic, the infection will grow exponentially again. We consider two different infection rates: 25% and 50% of the rate prior to the SaH order and three different reopening times–May 1, May 16, and June 1. Clearly, all of these scenarios lead to a substantial increase of infections and deaths, but we find that the infection rate is more critical than the timing of the reopening, pointing towards the importance of effective measures to reduce the infection rate after SaH orders are lifted. Besides lowering the infection rate, the growth can also be slowed by increasing the sum of the testing rate and the recovery rate of asymptomatic people. While the recovery rate is probably difficult to change, testing can be expanded. This highlights the importance of expanding testing capacity and encouraging early testing even without severe symptoms.

There are many deficits in the present model. Instead of a modified SIR model we could use a modified SEIR (Susceptible-Exposed-Infected-Recovered) model to better account for the incubation period. Additional improvement can be achieved by including other factors, such as correlation with different age groups, the availability of public health care, correlation with the health condition of the population, effects of the environment such as temperature and humidity, and many others. In particular, re-infection may be an important factor in the later stage of the epidemic.

Looking at less aggregated data, such as the data for each county or metropolitan area, might also be more meaningful than grouping the data by states which could include multiple metropolitan areas with different disease dynamics. It would be also interesting to study excess deaths instead of deaths directly produced by COVID-19. Additionally, many of the studied quantities are not expected to be Gaussian distributed. In the present work, we take either the mean or median values of their distributions. A more sophisticated study should include the distribution of these quantities to capture non-self-averaging effects. Similar to most approaches based on the SIR model, we implicitly assume that the population is homogeneous and well mixed, and that infection occurs without explicit time delay. The present model is essentially a mean field model with instantaneous coupling among different dynamical variables. It is worthwhile to have a detailed comparison between the present study and methods based on statistical inference of Gaussian-like distributions, such as the IHME model. [2] A simple comparison can be found in Figs 6 and 7.

After all, most studies of COVID-19 spread use highly cross-grained approximations. The detailed infection mechanism at the local level is largely ignored. A truly precise approach should include the dynamics of the interactions among people at the local level. For instance, it is clear that the dynamics in New York City cannot be the same as that at the rest of New York state. This difference, while important, is absent in all popular models being used for the study of the evolution of the COVID-19 pandemic. Utilizing a big data approach at the local level with graph theory should provide a more meaningful detailed analysis. Given the

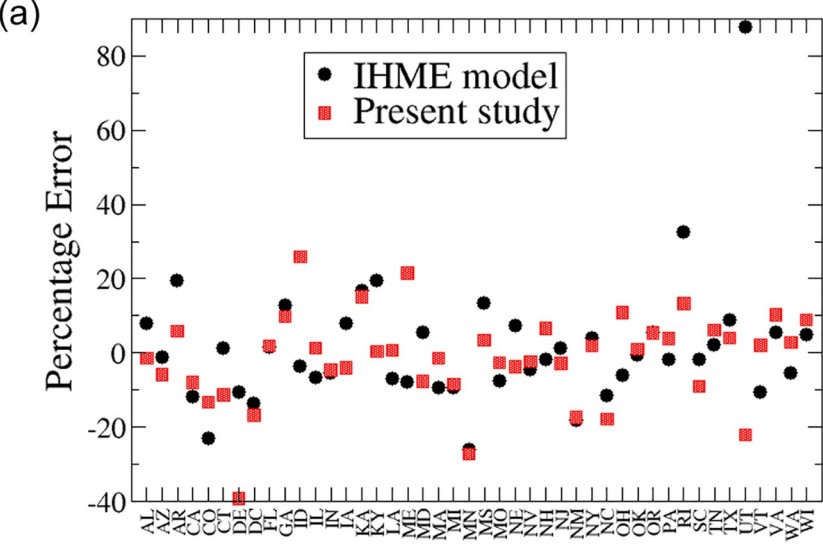

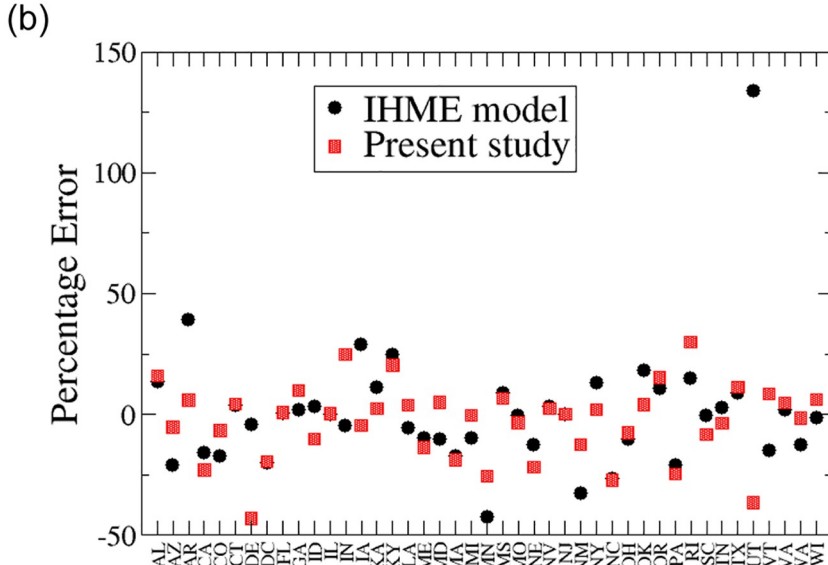

**Fig 6. Percentage error on the projected total death counts by April 27 of the IHME model and the present study for most states in the USA.** Positive and negative values respectively correspond to overestimates and underestimates. Note that we have subtracted 3,778 death counts from the IHME data for New York. Left panel: seven-day projection. Data from the IHME model is from its April 21 update. Data from the present study is generated from data prior to the same day. The average percentage error is 10.7% for the IHME model and 8.9% for our approach. Right panel: eleven-day projection. Percentage error corresponds to the projections based on data prior to April 17. The average percentage error is 15.2% for the IHME model and 11.7% for our model.

possibility of a second wave of infection, predictions at the local level may provide more focused mitigation approaches to minimize the economic and social impact of the pandemic.

Other important social and economic factors are missing from the present model, notably housing and household density, prevalence of multi-generational homes, rates of pre-existing conditions, poverty rates, and the fraction of the population with essential "higher risk" employment. These all suggest a less aggregated data should be considered for a more sophisticated modeling.

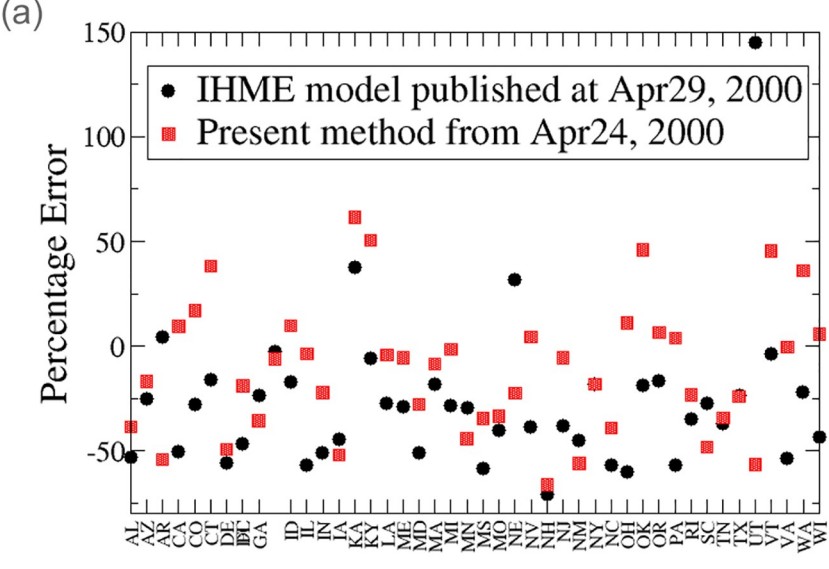

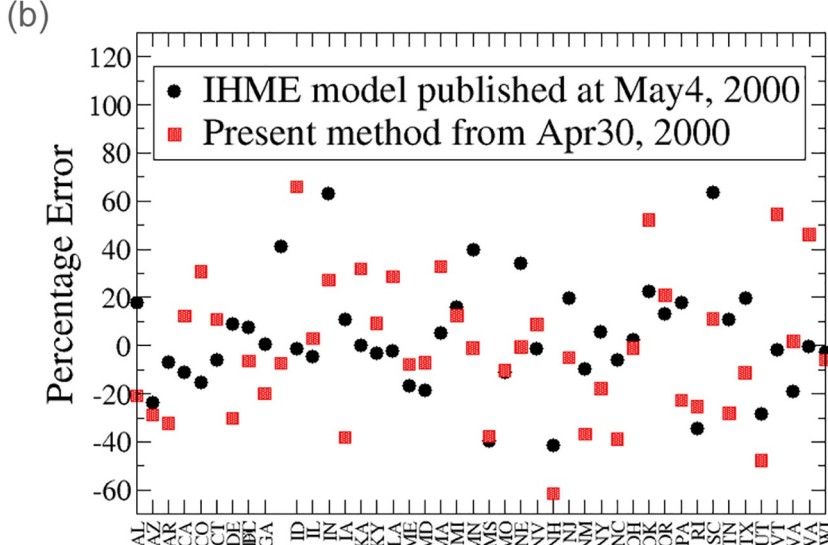

**Fig 7. Percentage error on the projected total death counts by May 31 from the IHME model and the present study for most states in the USA.** Positive and negative values correspond to overestimates and underestimates, respectively. Left panel: 38-day projection. Data from the IHME model is from its April 29 update. The average percentage error is 37.4% for the IHME model, and 27.2% for our approach. Right panel: 31-day projection. Percentage error corresponds to the projections based on data prior to May 4. The average percentage error is 16.5% for the new IHME model, and 23.0% for our approach.

In brief, perhaps the most timely information from this study is that reopening will definitely increase the projected number of cases and fatalities, but if the infection rate can be kept to a value much lower than the rate prior to the stay-at-home orders, exponential growth can be avoided. Control of the infection rate seems to be a more critical factor than the timing of the reopening. We have extended our approach to all states which had more than ten COVID-19 fatalities by April 10. Interested readers can find our predictions at https://covid19projection.org/, where we will update our projections in a timely manner. In particular, early estimates based on real data of the evolution of the spread after relaxing social

distancing measure will be an essential piece of information to predict and control the repercussions of the pandemic.

## Appendix: Benchmark against the IHME model

Here we compare our predictions with the ones from the IHME model [2] for seven- and eleven-day time intervals by calculating the percentage error. The percentage error is defined as:

$$\frac{\text{Model Projection} - \text{Actual Data}}{\text{Actual Data}} \times 100. \tag{17}$$

Since our method captures the effects of mitigation exclusively from the local data and these effects are not reflected in the data until around mid-April, both projections are for death counts on April 27.

For the seven-day projection, we take the April 21 update from the IHME model which includes data through April 20 [2] as well as our model results using the method described in the Method section with data up to the same day. We then compare the projected total death counts of both models with the data seven days later on April 27. Given the sparsity of the data, we do not expect meaningful results can be obtained for those states which recorded less than ten total death counts by April 10 within our model and they are not considered. These states include Alaska, Hawaii, Montana, North Dakota, South Dakota, West Virginia, and Wyoming. The left panel of Fig 6 displays the percentage error of the two models for the rest of the states in the USA. The average percentage error is 10.7% and 8.9% for the IHME model and our model, respectively.

Then, we repeat the comparison for a longer term projection of eleven days. We take the April 17 update from the IHME model, which includes the data up to April 16, [2] and our results generated with data up to the same day. We then compare the projected death counts of both models with the real data eleven days later on April 27. In this case, we also eliminate New Hampshire from the comparison since it did not record ten total deaths by April 6. The right panel of Fig 6 shows the percentage of error for both models. The average percentage error is 15.2% for the IHME model, and 11.7% for ours. We conclude that both model predictions are similar with our approach slightly outperforming the IHME model for short term projection. However, our projection is substantially higher than that of the IHME for longer term. A more thorough comparison is required to reveal the strengths and weaknesses of different models for simulating the spread of COVID-19.

## Further benchmark against the IHME model for longer term

We extend the comparison between our model and the IHME model for longer period of time. We updated our model with data up to April 24 and April 30. We compare our update for April 24 to that of the IHME model published at April 29. We also compare our update for April 30 to that of the IHME model published at May 4. We note that there is a substantial change in the IHME model from May 4, we denote it as new IHME model.

The left panel of Fig 7 displays the percentage error of our approach (from data up to April 24) and the old IHME model (from the update on April 29) on May 31. The right panel of Fig 7 displays the percentage error of our approach (from data up to April 30) and the new IHME model (from the update on May 4) on May 31. The new IHME model shows much higher infection and death counts than the old one. We choose to compare the predictions for May 31, as all states relax their mitigation efforts to different extents by the end of May. We expect

that the increasing social contact will affect the infection rate making our projection unreliable after the end of May, as we assume the mitigation efforts remain unchanged.

The average percentage error of the 38-day projection is 37.4% for the IHME model, and 27.2% for our approach. The average Percentage error of the 31-day projection is 16.5% for the new IHME model, and 23.0% for our approach. It shows that the new IHME model predictions have improved appreciably.

## Author Contributions

**Conceptualization:** Ka-Ming Tam, Nicholas Walker, Juana Moreno.

**Data curation:** Ka-Ming Tam.

**Formal analysis:** Juana Moreno.

**Investigation:** Ka-Ming Tam, Nicholas Walker, Juana Moreno.

**Methodology:** Ka-Ming Tam, Nicholas Walker, Juana Moreno.

**Writing – original draft:** Ka-Ming Tam, Nicholas Walker, Juana Moreno.

**Writing – review & editing:** Ka-Ming Tam, Nicholas Walker, Juana Moreno.

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
