## [Decision Letter · Decision Letter 0]

7 Aug 2020

PONE-D-20-14096

Effect of Mitigation Measures on the Spreading of COVID-19

in Hard-Hit States in the U.S.

PLOS ONE

Dear Dr. Ka Ming Tam,

Thank you for submitting your manuscript to PLOS ONE. After careful consideration, we feel that it has merit but does not fully meet PLOS ONE’s publication criteria as it currently stands. Therefore, we invite you to submit a revised version of the manuscript that addresses the points raised during the review process.

We look forward to receiving your revised manuscript.

Kind regards,

Francesco Di Gennaro

Academic Editor

PLOS ONE

Journal Requirements:

Additional Editor Comments (if provided):

Dear Authors,

I read with great pleasure your manuscript.

I believe that is a great article with high scientific impact

Below the reviewer's suggestions.

Reviewers' comments:

Reviewer's Responses to Questions

**Comments to the Author**

1. Is the manuscript technically sound, and do the data support the conclusions?

Reviewer #1: Partly

Reviewer #2: Yes

2. Has the statistical analysis been performed appropriately and rigorously? 

Reviewer #1: Yes

Reviewer #2: Yes

3. Have the authors made all data underlying the findings in their manuscript fully available?

Reviewer #1: Yes

Reviewer #2: Yes

4. Is the manuscript presented in an intelligible fashion and written in standard English?

Reviewer #1: Yes

Reviewer #2: Yes

5. Review Comments to the Author

Reviewer #1: I would like to thank the authors for this interesting work. Please find my detailed comments below:

Model:

There are considerable evidences that the COVID-19 recovered patients can be re-infected. For this reason, the recovered patients are considered to enter into the pool of susceptible population. As such, the SIR model could be a serious underestimation.

Any justification as to why SIR-S (susceptible-infected-recovered-susceptible again/re-infected) model was not considered?

Line 63-61: The authors should consider providing some brief details of the methods from their previous work here, so that this manuscript, if published, can also be a stand-alone paper. This will aid in the understanding of the general readers, who might or might not go and check the methods in their previous works. A detailed description is not necessary, just enough to assist the understanding of the results of this work.

Equation 9: (0) should be in subscript. For Equation 9, I see that the recovery rate for the asymptomatic cases were considered, then why wasn’t the recovery rate for quarantined people considered as well? Equation 9 would be substantially affected if the re-infected cases are considered as well. The authors should provide a strong justification as to why this could be overlooked.

Table 2: A percentage difference between the real and projected cases could be reported to help understand the deviation of the model from the reality.

I could not find any model validation and accuracy assessment for the projected model/figures. In lines 177-178, the IHME model was used to compare the results but would have been possible to start the projection from an earlier date and to carry out an accuracy assessment to understand the extent to which the projected figures are valid. Alternatively, the authors can now, in the revised version, consider validating the projected figures with the available data.

The results section is a bit difficult to follow, it would help if the authors can break it to subsections to facilitate a better understanding for the readers.

The discussion section was well written. However, what this study contributes in policy context (helping policy makers) should be explored too.

Reviewer #2: Really exceptional work and a robust model

Please proof read your article as there are several spelling and grammatical errors

Incredibly important factors are missing from the model, notably housing density, household density, multigenerational homes, poverty, rates of pre-existing conditions, % of population with essential "higher risk" employment - this should be called out as a limitation - it is hinted at by stating that less aggregated data by county should be considered, but a more deliberate inclusion of this limitation is warranted.

6. PLOS authors have the option to publish the peer review history of their article (what does this mean?). If published, this will include your full peer review and any attached files.

Reviewer #1: No

Reviewer #2: No

---

## [Author Response · Author response to Decision Letter 0]

10 Sep 2020

Reply to the Reviewers

Below we append a point-by-point response to the suggestions made by the reviewers. We would like to express our gratitude to the reviewers for their careful reading and helpful comments, which help us to improve our manuscript.

1 Reply to the first reviewer

1.1

Quote``There are considerable evidences that the COVID-19 recovered patients can be re-infected. For this reason, the recovered patients are considered to enter into the pool of susceptible population. As such, the SIR model could be a serious underestimation.

Any justification as to why SIR-S (susceptible-infected-recovered-susceptible again/re-infected) model was not considered?''

The referee is correct that there are considerable evidences that recovered COVID-19 patients can be re-infected. We agree that a SIR-S model provides a more complete approach for modeling such a situation. The reason we choose to ignore the contribution from re-infection is because the number of infected is still relatively small even today. For our work done last April, the total number of people who were ever infected in any state in the USA is probably less than 5% of the total population, and the reinfected count is a small percentage of the infected count. Thus, we choose to ignore the contribution of re-infected patients in our study. In the revised manuscript, we mention that re-infection may be an important factor in the later stage of the epidemic.

1.2

Quote``Line 63-61: The authors should consider providing some brief details of the methods from their previous work here, so that this manuscript, if published, can also be a stand-alone paper. This will aid in the understanding of the general readers, who might or might not go and check the methods in their previous works. A detailed description is not necessary, just enough to assist the understanding of the results of this work.''

We extended the description of our method in the revised manuscript and we believe the present manuscript is sufficiently self-contained such that the readers do not need to refer to our previous work.

1.3

Quote``Equation 9: (0) should be in subscript. For Equation 9, I see that the recovery rate for the asymptomatic cases were considered, then why wasn’t the recovery rate for quarantined people considered as well? Equation 9 would be substantially affected if the re-infected cases are considered as well. The authors should provide a strong justification as to why this could be overlooked.''

We defined I(0) as the unidentified infected population count at day 0, I(0) = I(t=0).

Eq. 9 must be understood together with Eq. 10, which approximates the count of identified quarantined patients at the start of the epidemic. We do not consider the reinfected count precisely because of our assumption that this number is much smaller than the number of infected. This assumption is particularly accurate for the very beginning of the epidemic, when Eqs. 9 and 10 are valid. Both equations 9 and 10 are only used to estimate the initial conditions of the very early stage of the epidemic. They are not valid at later stages, where the full set of equations 1-5 are needed for an accurate simulation of the pandemic.

We expanded the discussion of eqs. 9 and 10 in the revised manuscript.

1.4

Quote``Table 2: A percentage difference between the real and projected cases could be reported to help understand the deviation of the model from the reality. I could not find any model validation and accuracy assessment for the projected model/figures. In lines 177-178, the IHME model was used to compare the results but would have been possible to start the projection from an earlier date and to carry out an accuracy assessment to understand the extent to which the projected figures are valid. Alternatively, the authors can now, in the revised version, consider validating the projected figures with the available data.''

The validation and accuracy assessment for the projected model, including figures, has been provided in the appendix.

In the appendix of the previous manuscript, we have already compared the percentage difference between the real and projected cases for short term 7-day and 11-day projections. The percentage error is defined with respect to the real data. We write down this definition in the revised manuscript to clear possible confusion.

We extended the validation by considering 31-day (projection made at April 30, validation made at May 31) and 38-day (projection made at April 24, validation made at May 31) projections in the revised manuscript.

We chose to make the validation for May 31 as most states have relaxed their mitigation efforts to a certain degree by the end of May. We expect the infection rate is substantially increased after the relaxation of social distance measurements, as shown in the infection and death count data from June. The present data is calculated under the assumption that the mitigation efforts remain unchanged.

1.5

Quote``The results section is a bit difficult to follow, it would help if the authors can break it to subsections to facilitate a better understanding for the readers. The discussion section was well written. However, what this study contributes in policy context (helping policy makers) should be explored too.''

We broke down the results section into two subsections in the revised manuscript and we also extended the discussions on the results, which we believe has rendered the work more readable.

The main contribute in policy context is presented in the discussion section. The followings are quotes from the author summary section and the discussion section: ``In brief, perhaps the most timely information from this study is that reopening will definitely increase the projected number of cases and fatalities, but if the infection rate can be kept to a value much lower than the rate prior to the stay-at-home orders, exponential growth can be avoided. Control of the infection rate seems to be a more critical factor than the timing of the reopening."; ``we find that the infection rate is more critical than the timing of the reopening, pointing towards the importance of effective measures to reduce the infection rate after SaH orders are lifted. Besides lowering the infection rate, the growth can also be slowed by increasing the sum of the testing rate and the recovery rate of asymptomatic people. While the recovery rate is probably difficult to change, testing can be expanded. This highlights the importance of expanding testing capacity and encouraging early testing even without severe symptoms."

2 Reply to the second reviewer

2.1

Quote``Please proof read your article as there are several spelling and grammatical errors''

We thank the referee for pointing out the errors. We have proofread the manuscript and any spelling or grammatical errors should have been ironed out.

2.2

Quote``Incredibly important factors are missing from the model, notably housing density, household density, multigenerational homes, poverty, rates of pre-existing conditions, \\% of population with essential "higher risk" employment - this should be called out as a limitation - it is hinted at by stating that less aggregated data by county should be considered, but a more deliberate inclusion of this limitation is warranted.''

We thank the referee for pointing out these important factors. We have expanded our discussion to include these factors.

---

## [Decision Letter · Decision Letter 1]

6 Oct 2020

Effect of Mitigation Measures on the Spreading of COVID-19

in Hard-Hit States in the U.S.

PONE-D-20-14096R1

Dear Dr. Ming Tam,

We’re pleased to inform you that your manuscript has been judged scientifically suitable for publication and will be formally accepted for publication once it meets all outstanding technical requirements.

Kind regards,

Francesco Di Gennaro

Academic Editor

PLOS ONE

Additional Editor Comments (optional):

Dear Authors, congratulations!

Reviewers' comments:

Reviewer's Responses to Questions

**Comments to the Author**

1. If the authors have adequately addressed your comments raised in a previous round of review and you feel that this manuscript is now acceptable for publication, you may indicate that here to bypass the “Comments to the Author” section, enter your conflict of interest statement in the “Confidential to Editor” section, and submit your "Accept" recommendation.

Reviewer #1: All comments have been addressed

Reviewer #2: All comments have been addressed

2. Is the manuscript technically sound, and do the data support the conclusions?

Reviewer #1: Yes

Reviewer #2: Yes

3. Has the statistical analysis been performed appropriately and rigorously? 

Reviewer #1: Yes

Reviewer #2: Yes

4. Have the authors made all data underlying the findings in their manuscript fully available?

Reviewer #1: Yes

Reviewer #2: Yes

5. Is the manuscript presented in an intelligible fashion and written in standard English?

Reviewer #1: Yes

Reviewer #2: Yes

6. Review Comments to the Author

Reviewer #1: (No Response)

Reviewer #2: Thank you for addressing the noted concerns. Your responses clearly indicate your attention to the issues rasied and they are addressed in this version. Excellent manuscript and great contribution to the literature.

7. PLOS authors have the option to publish the peer review history of their article (what does this mean?). If published, this will include your full peer review and any attached files.

Reviewer #1: No

Reviewer #2: No

---

## [Editor Report · Acceptance letter]

15 Oct 2020

PONE-D-20-14096R1

Effect of Mitigation Measures on the Spreading of COVID-19 in Hard-Hit States in the U.S.

Dear Dr. Tam:

I'm pleased to inform you that your manuscript has been deemed suitable for publication in PLOS ONE. Congratulations! Your manuscript is now with our production department.

Kind regards,

on behalf of

Dr. Francesco Di Gennaro

Academic Editor

PLOS ONE